# Programme of triple-I mediator education (TIME) to improve medical disputes in clinical settings in Taiwan: a Delphi study

Yi-Chih Shiao  ,[1,2] Ruo-Nan Shen,[3] Wen-Wen Chen,[3] Yueh-Ping Liu,[4,5] Chung-Liang Shih,[5] Chih-Chia Wang[1]

[1]Department of Family and Community Medicine, Tri-Service General Hospital, National Defense Medical Center; School of Medicine, National Defense Medical Center, Taipei, Taiwan
[2]College of Law, National Chengchi University, Taipei, Taiwan
[3]Taiwan Drug Relief Foundation, Taipei, Taiwan
[4]Department of Medical Affairs, Ministry of Health and Welfare, Taipei, Taiwan
[5]Ministry of Health and Welfare, Taipei, Taiwan

**Correspondence to**
Dr Chih-Chia Wang; tsghccwang@gmail.com

## ABSTRACT

**Objectives** To establish a training programme to cultivate trainee mediation skills through time investment, skill incorporation and formation of in-house mediation services.

**Design** A four-round consensus conference was conducted by a number of seasoned experts selected in the manner of purposive sampling to determine core competences and relevant curricula through the modified Delphi process.

**Setting** Responses collected from enrolled experts through four rounds of the Delphi process from 11 November 2018 to 17 May 2019.

**Participants** Onboard seasoned mediators with different specialties.

**Outcome measures** Items with a median rating of 4 or more on a Likert scale of 1–7 points and 70% or more in agreement were identified as core competence and curricula.

**Results** Eleven enrolled experts reached the consensus about the training syllabus based on the 4-round agreement with four pillars of core competence, including 'knowledge base of law', 'internalisation of the denotative and connotative meanings of care', 'effective, smooth and timely communication' and 'conflict resolution'. To grasp the dynamics and diversity of medical disputes on target, it is necessary to have sufficient knowledge and skills. We arrange our course in the order of teaching materials with pure didactics in the former two and with mixed contents comprising lectures and field exercises in the rest two.

**Conclusions** The sample developed a syllabus to train apprentices to take intermediate responses to medical disputes through the skills of conflict resolution and establishment of effective communication to improve the relationship between patients/relatives and medical staff, as a result of eventually reducing the conversion rate from dispute into litigation or alternative pathway. Policymakers in healthcare and top management in healthcare institutions can use this syllabus to guide their future education and training programme.

## STRENGTHS AND LIMITATIONS OF THIS STUDY

⇒ The important strength of this study is the use of Delphi method providing anonymity to bring about detailed discussions.
⇒ This study is the first official training programme in Taiwan to train newcomers' skills in first-line mediation in clinical settings.
⇒ This study is the beginning of a 5-year quality improvement programme conducted by the Ministry of Healthcare and Welfare since 2018 and has the potential to be dedicated to policy improvement in fixing the broken and fragile relationships between medical personnel and patients/relatives.
⇒ Limitations of our study include potentially selection bias caused by the method of purposive sampling and a lack of a follow-up system to assess training results.

## INTRODUCTION

The number of medical disputes has been increasing.[1] An intermediate phase is the meaning of medical dispute between the preliminary status, which 'conflict' goes as a disagreement between medical staff and patients/relatives, and the dispute resolution status as the final stage. Most of these disputes belong to non-negligence after factual approach of causation in retrospective analysis all the time.[2] To resolve these disputes efficiently, several approaches have been used, including post-dispute problem-solving strategies with intrajuridical litigations and extrajuridical alternative dispute resolutions (ADRs).[3–5] ADR has been integrated into the healthcare system of many countries[6–8] for mitigating risk of being sued and making both patients and physicians satisfied.[3 4] Pre-dispute risk identification and early intervention, however, were merely merged into our current system.

As parts of dispute management, litigation and ADR, as post-dispute strategies, can be thought of as passive approaches compared with in-the-moment-of-dispute concerns from a third party independent from physicians and patients/relatives. In Taiwan, so-called

active concerns about disputes are provided by a number of experts to fix the breakage in relationships between medical staff and patients/relatives and are dedicated to reducing the ratios of dispute turning into litigation. These peace-bringing experts with different occupations, including physicians, lawyers, judges, psychologists and social workers, are invited by the Ministry of Healthcare and Welfare (MOHW) based on their contributions, such as frontline mediators in hospitals, volunteer counsellors in non-government organisations (NGOs), which are committed to the advocation of patient rights, and promoters of legal reform in the public and private sectors alike, so that our government can consult these experts when it encounters relevant disputes.

With the evolution of legislation, the draft act for the prevention and resolution of medical malpractice was proposed by the Executive Yuan in 2018. Under the legal framework, active dispute care can alleviate the high tensions in doctor–patient relationships strained by medical disputes. A great demand for caring talents is anticipated, leading to the start of an associated programme named the triple-I mediator education (TIME), authorised by the MOHW and characterised by 'in-house argument handling', 'immediate action' and 'integration resources', to fix this talent gap. The TIME programme is the first formal training programme, born in response to address prevention of dispute turning into litigation on the legal basis of Article 82 of the Medical Care Art and borrowed from the concept of medical emergency team[9][10] or rapid response team[11] for the purpose of high acuity response to early deterioration of relationships between patients and physicians.

TIME, as the beginning of a 5-year quality improvement series since 2018, a number of invited experts will attend a consensus meeting, conducted in a four-round Delphi process, for the extraction of experts' experiences as the base of constructing a syllabus, including teaching knowledge and skills in the mediation field, to educate newcomers to become competent mediators. The study aims to identify the core competencies and confirm the teaching materials for future tutors to provide precise instructions to trainees.

## METHODS

### Patient and public involvement
No patients and public were not involved in this study.

### The Delphi method
The Delphi method is reliable and has been effectively used for establishing consensus regarding healthcare management.[12] It is a multistage, iterative and anonymous process consisting of at least two rounds of surveys.[13] Opinions and feedback were structurally categorised under different headings after each round and exported to a temporary meeting document as reference to effectively facilitate discussions among moderators in the next round. Considering the critical paucity of relevant evidence for fostering skills, the Delphi method with four rounds of surveys was introduced to achieve consensus regarding the syllabus for the TIME programme. The strengths of this method included anonymity, capability of obtaining a collection of ideas and opinions from participants, and cost effectiveness. The weaknesses were the length of time required, research bias through the potential influence of moderators or participants and uneven distribution of specialties among the respondents. The authors followed the guidance of Conducting and REporting DElphi Studies.[14]

### Agreement and consensus
Agreement was defined as the percentage of positive rather than neutral or negative responses divided by the total completed responses. Consensus was defined as 70% or more agreement according to the recommendations made by Sumsion.[15]

### The moderator of the time programme
To integrate experts' opinions effectively, a clinical physician practising family medicine with position as an associate professor in medicine and a Doctor of Laws degree is designated as the moderator by the Taiwan Drug Relief Foundation (TDRF) due to his multiple experiences for interdisciplinary coordination, such as conducting a pilot project to encourage medical institutions to properly handle surgical and anaesthesia disputes in 2014 and bringing about the satisfactory outcome of the promotion of the Childbirth Accident Emergency Relief Act in 2015.

### The sample of experts in the time program
Delbecq et al suggested that 10–15 members could be a satisfactory range for sample.[16] Our goal was to have at least 10 participants for each round. The experts participating in the TIME programme were formally recruited in the manner of purposive sampling through granted consent of the expert invitation document sent by TDRF. These experts had more than 10 years of experience making sufficient contributions in the field of medical personnel–patient relationship repair, including serving as local civil mediators in various courts affiliated to the Judicial Yuan, and NGOs that provided legal services to patients/family members. Although the number of physicians outweighed other types of profession, it did not mean they stand the physicians' point of view. Instead, two of them provided long-term mental support and counselling to patients who were stuck in a poor relationship with their medical healthcare providers. In order to make sure a whole coverage of experts' opinions, the common triad encountered in medical disputes, including the arm of medical personnel, the arm of patients/relatives and the arm of legal professionals, was included in our study.

### Rapid reviews
'Rapid reviews' is a useful tool accepted by the WHO to furnish evidence in the target field in a cost-effective fashion.[17] References with respect to dispute resolving and relevant training programmes, reviewed by the

moderator, would be provided to the participants in the first Delphi round. The spectrum of keywords would be set up as wide as possible to include enough publications in the field of training courses for resolving medical disputes (ie, 'medical dispute', 'conflict' and 'training'). Articles published between 1998 and 2018, written in English and indexed in common databases, such as PubMed, Medline and Embase, would be included. The sample considered whether they exhibited pillars of professionalism in the TIME programme based on relevant content in medical education,[18] Roach's six C's in caring[19] and a communication model called CI-CARE proposed by the University of California, Los Angeles Health System,[20] as a benchmark. Furthermore, evidence unveiled mediation training could facilitate healthcare staff to resolve conflicts at an early stage.[21] The basics of training should include several aspects of conflicts, comprising the identification of conflict resources, application of conflict management and upgrading non-verbal communication skills.[22] A follow-up interval of 6 months after the training programme showed that the trainees became more competent in coping with conflicts.[23]

### Delphi survey rounds

In round 1, the sample was required to read aforementioned references reviewed by the moderator and provided details of their occupations, experience with regard to dealing with medical disputes in clinical settings, knowledge of conflict management and coping strategies used to alleviate the high tensions between medical personnel and patients/relatives. Opinions and details on constructing core competences correlated with fostering professionalism and reflection after reading references were provided using narrative text responses and discussion. The feedback and views of experts were carefully gathered to generate a draft questionnaire.

In round 2, the draft proposal comprising several features of core competences was presented and measured through a questionnaire answered using a 7-point Likert-type scale (1=strongly disagree, 2=disagree, 3=somewhat disagree, 4=neutral, 5=somewhat agree, 6=agree and 7=strongly agree). Consensus was based on an agreement level of >70%. Items with <70% agreement and relevant feedback regarding disagreement with a core competence was discussed by the moderator group. Participants were then requested to share opinions and details for constructing the core curricula correlating to fostering core competences by using narrative text responses. Information on positive and negative aspects mentioned earlier was gathered systematically to generate the initial draft containing the core competences and core curricula of the TIME programme.

In round 3, agreement among the panel members regarding the draft of the core curricula was obtained using a 7-point Likert scale similar to that used in round 2 with respect to the level of agreement by the respondents. Consensus was reached and discussions were held by the moderator group if any item had <70% agreement.

Additional narrative text was provided to determine whether any disagreement existed regarding the core curriculum. This draft was revised by the group based on the analysis of agreement and annotation of round 3. Besides, participants were asked to calculate the time in each segment of this training programme in self-simulated situations and the scheduled class time would be discussed in the fourth round.

In round 4, respondents were asked to provide suggestions on syntactic errors, improper phrasing or terminology to make the description concise. The final version of the TIME training syllabus included (1) immediate action, (2) policy incorporation and (3) in-house settlement based on agreement in rounds 2 and 3. In regard to the class, time of each segment, including 'knowledge base of law', 'internalisation of the denotative and connotative meanings of care', 'effective, smooth and timely communication' and 'conflict resolution', would be 1 hour, 1 hour, 3 hours and 3 hours, respectively, after the participants reached the consensus after taking the reasonable length of content in each segment of this course into accounts. The final version of the syllabus draft was validated and announced on 15 November 2019, the promulgation date (figure 1), accompanied with the announcement of the initial version of the official course handbook based on the draft by the MOHW. The latest version on the official website of the MOHW was free to the public with open access.[24]

### RESULTS

The characteristics of the participants are presented in table 1. All the 11 participants completed the 4-round Delphi survey. The expert panel group consisted of 7 men (63.6%) and 4 women (36.4%). Physicians predominated the expert panel group (54.5 %). The respondents enrolled were experienced, knowledgeable and skilful specialists of different fields possessing more than 10-year experience.

Table 2 shows a consensus on all items, with more than 70% agreement. The item 'effective, smooth and timely communication' had the highest number of responses of 'strongly agree' (n=10, 90.9%), and the rest of the items, including 'knowledge base of law', 'internalisation of the denotative and connotative meanings of care', and 'conflict resolution', had almost equal agreement (n=6, 54.5%). Only the item 'knowledge base of the law' elicited a neutral opinion (n=2, 18.2%).

In table 3, a consensus was reached for all items, with an agreement of 100%, except for the item 'proceeding in law case with medical disputes: conventional means', which had 91.7% agreement. Only in the 'knowledge base of law' category was the number of participants who responded of strongly agree below 10 (n=9). The other three categories had at least one item with >10 people indicating strong agreement. Additionally, in the categories of 'effective, smooth and timely communication' and 'conflict resolution', consensus was reached on all of the correlated core

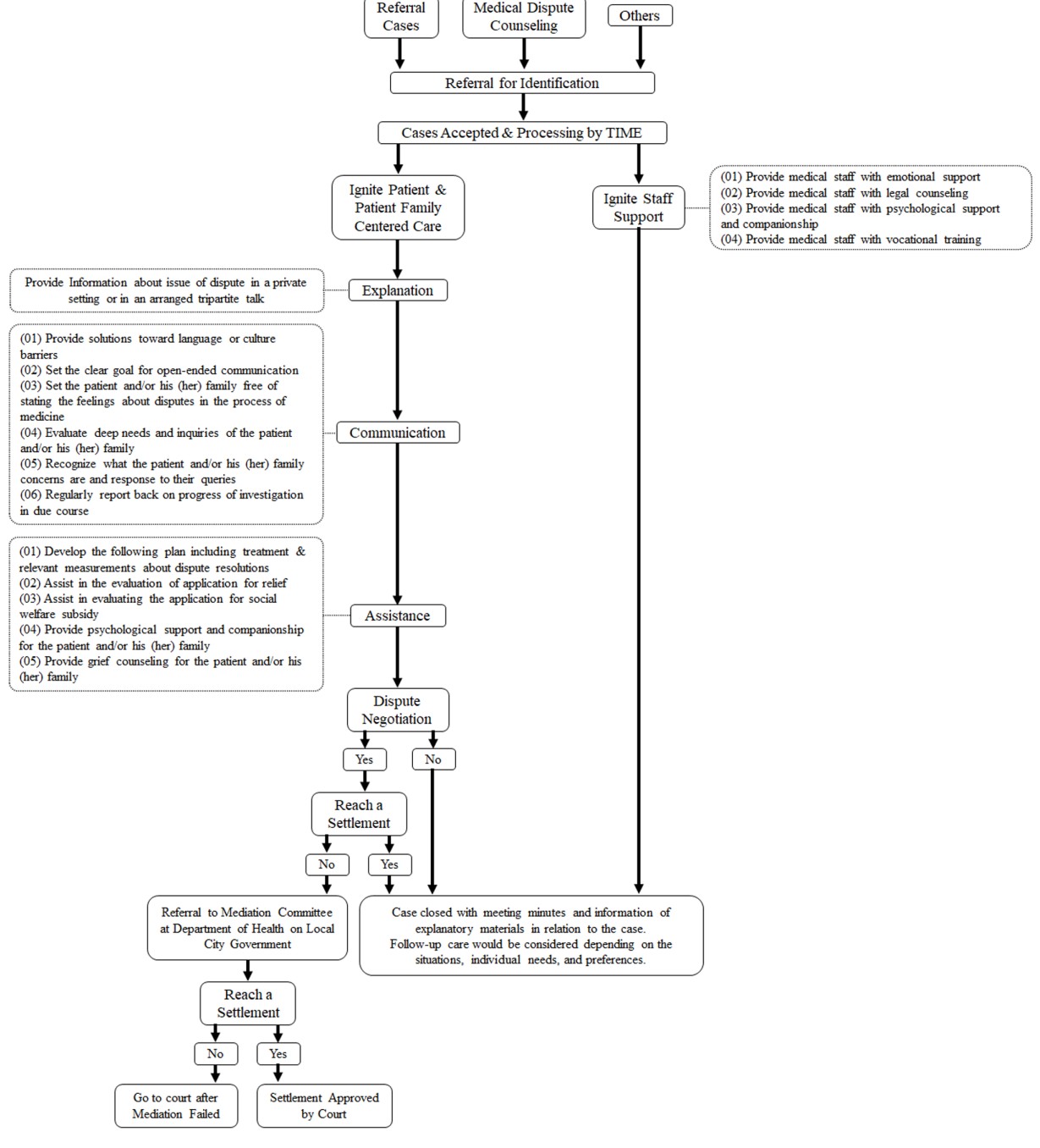

**Figure 1** Flowchart of the execution of the triple-I mediator education programme (TIME). If a dispute case is referred for dispute consultation, mediators attend the relevant location and collaborate with their team members of different specialties depending on the scale and type of the dispute. As a fair and objective third party, they provide their services equally to both patients and medical staff. If a settlement is not reached, the dispute case is referred to a mediation committee at the department of health of the local city government for further case review and processing. If both medical staff and patients cannot reach a final agreement, the case is sent for a trial for a final decision to be made by a judge.

curricula, with more than half of the members in strong agreement. Furthermore, only the category of 'effective, smooth and timely communication' had two items for which 10 members were in strong agreement, namely, 'ethics of

TIME: knowing the difference between what you have a right to do and what is right to do' and 'establishment of communication: an impregnable foundation leaving no boundary and obstacle'.

**Table 1** Characteristics of the panel members who completed the 4-round Delphi survey

|  | Number (%) |
|---|---|
| **Sex** | |
| Male | 7 (63.6) |
| Female | 4 (36.4) |
| **Age** | |
| ≥50 years | 8 (72.7) |
| <50 years | 3 (27.3) |
| **Occupation** | |
| Physician | 6 (54.5) |
| Lawyer | 2 (18.2) |
| Judge | 1 (9.1) |
| Clinical psychologist | 1 (9.1) |
| Social worker | 1 (9.1) |
| **Experience with mediation, years** | |
| ≥10 years | 11 (100) |
| <10 years | 0 |

## DISCUSSION

After a four-round Delphi process was applied, consensus was reached regarding the syllabus for the TIME training programme, which was established by a panel consisting of highly experienced experts in mediation. Table 4 was the final product containing four items on core competencies with respect to relevant core curricula. Each item received at least 70% agreement, which represents convergence of opinions.

To optimise the present healthcare system, Taiwan's MOHW introduced an in-house mediation (IHM) system for conflict management in 2013.[25] According to current regulations on IHM in Taiwan, mediators are defined as sentinel and people with an engaging personality[26] who provide mediation for a suspected, gradually formed or established dispute, and collaborate with team members from different specialties depending on the scale or type of dispute. The mediator is similar with a symphony conductor, setting the tempo and guaranteeing proper entries for the ensemble cast,[27] to put fragment of narratives of the conflict together into euphony instead of cacophony. Of the total number of members in the team,

at least three must be in-house medical staff in addition to the original teams of healthcare providers and non-medical staff (eg, legal professionals or social workers) and non-medical staff should comprise at least one-third of the team in line with law. The design of IHM is aimed at creating a 360° feedback mechanism and obtaining good understanding of excellence; in reality, the ultimate goal of IHM application in dispute resolution is to ensure highly effective teamwork for appropriate dispute resolution rather than blame shifting.

The quality of communication can determine the direction of the relationships between medical staff and patients/relatives. Poor communication stems from the gap in mutual understanding, moving to incredulity[28] and causing the overuse of defensive medicine as a nonviolent strategy in the management of medical disputes.[29] This psychological chain of reaction could be fixed from the technical perspective of resolution management, such as using Google Translate, a famous applied science-oriented technology, helping overcome language barriers and improve satisfaction levels among healthcare providers and patients.[30] The design of ideal mediators after our TIME training is not focused on technical aspects mentioned before but on in-person communication by arriving at the scene of the dispute immediately and providing immediate care. In hospital settings, a mediator can activate the integration of resources after engaging with both parties of a dispute. In clinical settings, patients focus on discomfort and ignore or miss relevant information regarding available treatments. The utility of this related mediation may be time sensitive. Mediators accelerate the use of individualised services targeting specific demands after the overall evaluation of case needs and concerns. For those with legal liability for any damage they cause, legal remedies affirm the validity of their claims and requests. Damage caused by those with no liability could be dealt with through compensation.[31] Common access points to humanitarian relief include Taiwan's drug injury relief system, compensation for vaccine injuries and childbirth accident relief.

The core concept conveyed in the segment 'effective, smooth and timely communication' is that the quality of care would be maintained by the high quality of patient-centred caring (PCC).[32 33] As the first step of PCC, demonstrating empathy, including verbal and non-verbal (eg, body expressions[34]) clues in every visit, is a critical

**Table 2** Round 2 survey responses: core competences of mediators

|  | Strongly agree (n, %) | Agree (n, %) | Somewhat agree (n, %) | Neutral (n, %) | Somewhat disagree (n, %) | Disagree (n, %) | Strongly disagree (n, %) | Agreement (n, %) |
|---|---|---|---|---|---|---|---|---|
| Knowledge base of law | 6 (54.5) | 1 (9.1) | 2 (18.2) | 2 (18.2) | 0 (0) | 0 (0) | 0 (0) | 9 (81.2%) |
| Internalisation of the denotative and connotative meanings of care | 6 (54.5) | 4 (36.4) | 1 (9.1) | 0 (0) | 0 (0) | 0 (0) | 0 (0) | 11 (100%) |
| Effective, smooth and timely communication | 10 (90.9) | 1 (9.1) | 0 (0) | 0 (0) | 0 (0) | 0 (0) | 0 (0) | 11 (100%) |
| Conflict resolution | 6 (54.5) | 5 (45.5) | 0 (0) | 0 (0) | 0 (0) | 0 (0) | 0 (0) | 11 (100%) |

**Table 3** Round 3 survey responses: core curricula of mediators

| Core competence | Core curriculum | Strongly agree (n, %) | Agree (n, %) | Somewhat agree (n, %) | Neutral (n, %) | Somewhat disagree (n, %) | Disagree (n, %) | Strongly disagree (n, %) | Agreement (n, %) |
|---|---|---|---|---|---|---|---|---|---|
| Knowledge base of law | Overview of medical disputes and medical malpractice: current trends in Taiwan | 4 (36.3) | 5 (45.5) | 2 (18.2) | 0 (0) | 0 (0) | 0 (0) | 0 (0) | 11 (100%) |
| | Tort law dealing with professional negligence in medical malpractice and tort liability related to medical accidents: rights and responsibilities you should know | 5 (45.5) | 4 (36.3) | 2 (18.2) | 0 (0) | 0 (0) | 0 (0) | 0 (0) | 11 (100%) |
| | Proceedings in a law case with medical disputes: conventional means | 3 (27.3) | 3 (27.3) | 4 (36.3) | 1 (9.1) | 0 (0) | 0 (0) | 0 (0) | 90.9% |
| | Current non-contentious approaches in dealing with dispute resolution: approaches involving tactical flexibility | 7 (63.6) | 4 (36.4) | 0 (0) | 0 (0) | 0 (0) | 0 (0) | 0 (0) | 11 (100%) |
| Internalisation of the denotative and connotative meanings of care | Core meanings of person-centred care: guidance involving a holistic approach | 9 (81.8) | 2 (18.2) | 0 (0) | 0 (0) | 0 (0) | 0 (0) | 0 (0) | 11 (100%) |
| | Practical aspects of person-centred care: be straightforward without being rude | 10 (90.9) | 1 (9.1) | 0 (0) | 0 (0) | 0 (0) | 0 (0) | 0 (0) | 11 (100%) |
| | Duties and responsibilities of a triple-I health mediator: balancing obligation and liability | 6 (54.5) | 5 (45.5) | 0 (0) | 0 (0) | 0 (0) | 0 (0) | 0 (0) | 11 (100%) |
| | Nationwide accessibility of mediation services: barriers no longer exist among different levels of healthcare institutions | 4 (36.4) | 7 (63.6) | 0 (0) | 0 (0) | 0 (0) | 0 (0) | 0 (0) | 11 (100%) |
| | Integration of present-day resources: foundations dealing with drug relief, childbirth accident relief, damage compensation for injury and programme for national vaccine injury compensation | 4 (36.4) | 4 (36.4) | 3 (27.2) | 0 (0) | 0 (0) | 0 (0) | 0 (0) | 11 (100%) |
| Effective, smooth and timely communication | Ethics of a triple-I health mediator: knowing the difference between what you have a right to do and what is right to do | 10 (90.9) | 1 (9.1) | 0 (0) | 0 (0) | 0 (0) | 0 (0) | 0 (0) | 11 (100%) |
| | Predecessors in the management of mediation: think ahead in time | 9 (81.8) | 2 (18.2) | 0 (0) | 0 (0) | 0 (0) | 0 (0) | 0 (0) | 11 (100%) |
| | Establishment of communication: an impregnable foundation leaving no boundary and obstacle | 10 (90.9) | 1 (9.1) | 0 (0) | 0 (0) | 0 (0) | 0 (0) | 0 (0) | 11 (100%) |
| | Acknowledge the needs and expectations of patients/relatives and medical personnel | 8 (72.7) | 3 (27.3) | 0 (0) | 0 (0) | 0 (0) | 0 (0) | 0 (0) | 11 (100%) |
| | Psychological analysis of patients: critical starting point of lean thinking | 8 (72.7) | 3 (27.3) | 0 (0) | 0 (0) | 0 (0) | 0 (0) | 0 (0) | 11 (100%) |
| | Grief counselling: reach out for help | 7 (63.6) | 4 (36.4) | 0 (0) | 0 (0) | 0 (0) | 0 (0) | 0 (0) | 11 (100%) |
| | Emotion recognition and corresponding responses: knowing me, knowing you | 9 (81.8) | 2 (18.2) | 0 (0) | 0 (0) | 0 (0) | 0 (0) | 0 (0) | 11 (100%) |

Continued

**Table 3** Continued

| Core competence | Core curriculum | Strongly agree (n, %) | Agree (n, %) | Somewhat agree (n, %) | Neutral (n, %) | Somewhat disagree (n, %) | Disagree (n, %) | Strongly disagree (n, %) | Agreement (n, %) |
|---|---|---|---|---|---|---|---|---|---|
| Conflict resolution | Disparity in problem recognition between physicians and patients: gaps you cannot ignore | 6 (54.5) | 5 (45.5) | 0 (0) | 0 (0) | 0 (0) | 0 (0) | 0 (0) | 11 (100%) |
| | Strategies and tips for conflict resolution: learn the ropes in advance | 9 (81.8) | 2 (18.2) | 0 (0) | 0 (0) | 0 (0) | 0 (0) | 0 (0) | 11 (100%) |
| | Tactics and hints for facilitating effective communication: the bond-forming factor in all human relations is conversation | 8 (72.7) | 3 (27.3) | 0 (0) | 0 (0) | 0 (0) | 0 (0) | 0 (0) | 11 (100%) |
| | Frequently asked questions and answers: power-up your abilities without hesitation | 10 (90.9) | 1 (9.1) | 0 (0) | 0 (0) | 0 (0) | 0 (0) | 0 (0) | 11 (100%) |

and teachable multiphrase skill.[35] Six keys are taught in our course to facilitate effective empathy,[36] including (1) discerning existence of emotions in the clinical setting, (2) taking a pause to imagine what the interviewer (eg, medical personnel or patients/relatives) may think about, (3) stating mediators' perception of the interviewer's feeling (ie, 'It sounds like this event really frustrates you in depth … '), (4) making the stated feeling presented by the interviewer plausible, (5) showing respect for the interviewer's effort to tackle the dilemma and (6) furnishing mental support and partnership (ie, 'Let's move to the action that we can take together to … '). The trainers would teach trainees a formula comprising queries clarifications—in order to memorise and to practice.[37]

Trainees would learn the ropes of integrating emotional clues about physical or mental health from tutors in our teaching programme. These collected threads will be the base for approaching the interviewees involved in medical dispute authentic feelings. To be more specific, detected emotions could be approximately categorised in the Kübler-Ross Change Curve (KRCC), including grief denial, anger, bargaining, depression, acceptance and coping strategies, could be conducted according to individual differences. However, one thing should be kept in mind that this category is just an auxiliary measure for approaching the interviewer's authentic feelings. Stiffly putting someone's fragment of emotional journey into the framework of the KRCC is inappropriate.[38] These detections of emotions by a well-educated mediator facilitate precise and rapid responses to such feelings because physical discomfort leads to mental and emotional suffering.[39] As the status of physical health decline, either temporarily or permanently, would have mutual[40] and detrimental effects on psychological health (eg, anxiety[41] and depression[42]), leading to eventually undermining the existing/established-in-the-process partnerships and difficulty in building trust with others.[43] The 2-hour practice in this segment of the TIME training programme is designed to allow trainees to be proficient in providing emotional support through situational exercise and to receive verbal feedback on their performances by trainers.

When a medical dispute occurs, both medical personnel[44] and patients/relatives[45] would be influenced and become vulnerable, sensitive and self-absorbed. The defence mechanism will be activated in both parties and cause the relationship to become increasingly alienated and destructive, and make people more injured, bringing about a form of vicious cycle. To break this cycle and restore the broken trust, the so-called 'transformative mediation'[46] with two key features is fundamental and would be practiced in the practicing part of the course section 'Conflict resolution'. One is empowerment shift,[47] which could allow both parties to regain respect, confidence and empathy. The other is recognition shift,[47] which could help patients reduce their defensiveness against medical staff and gain a different perspective. This approach made patients express needs and strengthened bilateral participation in healthcare issue.[48]

**Table 4** Final product of core curricula after a 4-round Delphi survey

| Curriculum | Duration | Description |
|---|---|---|
| Knowledge base of law | 1 hour | (1) Overview of medical dispute and medical malpractice: currents trend in Taiwan; (2) tort law dealing with professional negligence in medical malpractice and tort liability related to medical accidents: rights and responsibilities you should know; (3) proceedings in a law case with medical disputes: conventional means; (4) current non-contentious approaches in dealing with dispute resolution: approaches involving tactical flexibility |
| Internalisation of the denotative and connotative meanings of care | 1 hour | (1) core meanings of person-centred care: guidance involving holistic approach; (2) practical aspects of person-centred care: be straightforward without being rude; (3) duties and responsibilities of a triple-I health mediator: balancing obligation and liability; (4) nationwide accessibility of mediation services: barriers no longer exist among different levels of healthcare institutions; (5) integration of present-day resources: foundations dealing with drug relief, childbirth accident relief, damage compensation for injury and programme for national vaccine injury compensations |
| Effective, smooth and timely communication (1-hour lectures, 2-hour practice) | 3 hours | (1) Ethics of a triple-I health mediator: knowing the difference between what you have a right to do and what is right to do; (2) predecessors in the management of mediation: think ahead in time; (3) establishment of communication: an impregnable foundation leaving no boundary and obstacle; (4) acknowledge the needs and expectations of patients/relatives and medical personnel; (5) psychological analysis of patients: critical starting point of lean thinking; (6) grief counselling: reach out for help; (7) emotion recognition and corresponding responses: knowing me, knowing you |
| Conflict resolution (1-hour lectures, 2-hour practice) | 3 hours | (1) Disparity in problem recognition between physicians and patients: gaps you cannot ignore; (2) strategies and tips for conflict resolution: learn the ropes in advance; (3) tactics and hints for facilitating effective communication: the bond-forming factor in all human relations is conversation; (4) frequently asked questions and answers: power-up your abilities without hesitation |

As we focused on the arm of patients/relatives, there would be ignorance of the need for care from the arm of medical personnel. This constantly neglected blind spot of calling for help for healthcare professions is going to be smoothened by mediators after the TIME training. The potential emotional trauma of staff in response to adverse patient responses can cause burnout.[49] Such burnout has been thought to be high among physicians globally.[50] Consequences of burnout have detrimental effects on doctors, which may lead to them making incorrect decisions (thus causing medical errors), exhibiting a hostile attitude toward patients, and having dysfunctional relationships with colleagues.[49] Once staff having traumatic experiences, they would suffer from second victim syndrome (SVS).[51] Mediators after the TIME training could recognise the six phrases, including the early stage of 'initial chaos and accident response' and the final of 'moving on'.[52] In half of the process during the SVS recovery trajectory, especially in phrase three, the suffering personnel would try to seek help and re-establish trust with surroundings.[52] This would be the timing on target for mediators to approach involved personnel by listening empathetically, eschew judgement and showing respect.[53] SVS cannot be treated if systematic and well-structured support for coping with emotional burdens is absent.[54] To put it in a nutshell, medical staff should be provided with adequate care similar to support for patients/relatives with mature, intact and positive counselling.[55]

Some limitations must be noted. First, the sample composition was imbalanced due to an uneven distribution of specialties caused by the method of purposive sampling. Doubts arise as any possibility of selection bias still left. Second, a retrospective follow-up to assess training results was lacking. To understand the quality of the mediation service under current circumstances, establishment of a distinctive measurement tool with indicators for capabilities of perception, accountabilities, communication and assistance is recommended. Third, a feedback mechanism should set up for the collection of opinions and suggestions from onboard mediators, specialists taking part in any step of the mediation or litigation, or our greenhorns to revise the curriculum. This syllabus could be considered a starting point, but further research is required to achieve perfection.

## CONCLUSIONS

The TIME syllabus is the first official training programme of mediators for medical disputes in Taiwan and is constructed after a four-round Delphi process with four sections, including 'knowledge base of law', 'internalisation of the denotative and connotative meanings of care', 'effective, smooth and timely communication' and 'conflict resolution'. The design of the TIME syllabus aims to enrich trainees the knowledge in relation to mediation and sharpen their skills in establishing firm relationships with medical personnel and/or patients/relatives and managing medical disputes through the application of lectures and field exercises.

**Acknowledgements** The authors thank Professor Che-Ming Yang for sharing their experiences and providing input regarding mediation during medical disputes and Yu-Ying Huang (Attorney at Law; Chairman of Taiwan Association of Harmony Medicare and Patient) for legal counselling. The authors appreciate the contribution of specialists from the Taiwan Ministry of Health and Welfare (registration number: MOHW1100023690) and the ministry of Science and Technology, Taiwan (Grant No: MOST 110-2511-H-016-002-MY3) in the integration of knowledge. The authors also thank the Taiwan Drug Relief Foundation's project working group members, including Pin-Hsien Yeh, Hsin-Hsin Chen and Po-Han Chen, for their contributions toward data collection and analysis and administrative support. We acknowledge the editor and series editor of *BMJ Open* for constructive criticism of an earlier version of this article.

**Contributors** YCS contributed to literature review, plan execution, data collection and management, and manuscript draft preparation. RNS contributed to the study design, data collection and management, and assistance to the principal investigator in programme execution. WWC was the current president of Taiwan Drug Relief Foundation and contributed to policy advisory and study design. YPL and CLS contributed to policy advisory and information gathering to develop, support and implement policies. CCW is the guarantor of the study and the principal investigator and contributed to the study conceptualisation and design, data analysis evaluation and programme execution. The final manuscript was read and approved by all authors. The corresponding author attests that all listed authors meet authorship criteria and that no others meeting the criteria have been omitted.

**Funding** This work was supported by Taiwan's Ministry of Health and Welfare and the ministry of Science and Technology (Grant No: MOST 110-2511-H-016-002-MY3) on the basis of the 2018 Quality Improvement Programme in Medical Dispute Resolution (grant number: M07A3401).

**Competing interests** None declared.

**Patient and public involvement** Patients and/or the public were not involved in the design, or conduct, or reporting, or dissemination plans of this research.

**Patient consent for publication** Not applicable.

**Ethics approval** This study involves human participants. The triple-I mediator education programme was approved by the institutional review board and the Research Ethics Committee of the Tri-Service General Hospital (reference number: E202216022) and was conducted in line with the Declaration of Helsinki. Informed consent was obtained along with the participants' responses in the study before taking part.

**Provenance and peer review** Not commissioned; externally peer reviewed.

**Data availability statement** Data are available upon reasonable request.

**ORCID iD**
Yi-Chih Shiao http://orcid.org/0000-0002-0100-6004

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
