## [Reviewer comments · BMJ Open]

ARTICLE DETAILS

TITLE (PROVISIONAL)	The program of Triple-I mediator education (TIME) to improve medical disputes in clinical settings in Taiwan: a Delphi study
AUTHORS	Shiao, Yi-Chih; Shen, Ruo-Nan; Chen, Wen-Wen; Liu, Yueh-Ping; SHIH, Chung-Liang; Wang, Chih-Chia

VERSION 1 – REVIEW

REVIEWER	Piryani, Rano Mal Universal College of Medical Sciences, Department of Internal Medicine
REVIEW RETURNED	30-Nov-2021

GENERAL COMMENTS	It seems authors have contributed their maximum efforts in conducting this study and writing this manuscript.
---

REVIEWER	Lin , Jin-Ding Mackay Medical College, Institute of Long Term Care
REVIEW RETURNED	31-Jan-2022

GENERAL COMMENTS	1.The necessity of this study need to emphasize to increase the importance in this study.2. The result of the abstract should have deep description instead of general information to the competence.3. The author should have a brief description of previous training or teaching method, particularly in its pros and cons.4. The panel group was chosen of approved by the Taiwan's MOHW, is it have selection bias? it should have a discussion.5. The effect of the panel member occupation toward this study should have a discussion in the content.
--

REVIEWER	Forbat, Liz University of Stirling
REVIEW RETURNED	23-May-2022

GENERAL COMMENTS	I found this paper had promise and detailing the provenance of a training programme in medical conflict/dispute management has a lot to offer. There were a number of areas in the paper which require tightening up however, as detailed below. Title: add in Delphi method and the purpose of the paper, to indicate that this paper is about developing an educational intervention via a specific research method. Abstract: include sample size. Conclusion – is it just between staff and patient? There is a large body of work on conflict in pediatric settings which is framed around the triad dynamic, i.e the position of a relative into the conflict, which is very common and can
--

increase the potential for medical disputes when the patient cannot advocate for themselves.

Strengths: perhaps the language 'panel group' would be better replaced with the idea of 'sample'. Reconsider 'naive trainers' as this phrase isn't easily understandable without further context.

Introduction

Can you define 'medical dispute'? is this different to conflict? Or litigation? Is it only between staff and patients, or intra-staff or intra-organisational conflict too?

Since TIME is defined as about dispute prevention, how does this sit alongside ideas of dispute management/resolution? They are on a continuum but if one is about prevention and one about resolution then they need to be conceptually presented clearly.

Can you provide more information on what TIME task force are 'seasoned experts' in – what do they do, how many are there, what training have they had, how are they selected, what evaluations have been conducted? You mention there's no systematic course 'in the long run' but it isn't clear what this means – is there a short course they do?

I was expecting to see a brief section detailing what other trainings are available for managing disputes in medical settings, internationally. One such approach has not been cited, but there may be others. <https://pubmed.ncbi.nlm.nih.gov/27098546/> This is especially important given that this was part of your methods process as stated on p7.

Methods

can you explain why patients/public were not involved? Since medical disputes are framed in the introduction as between health care staff and patients (i.e. not between healthcare staff) then it might have been useful to integrate their perspectives.

Did you use the CREDES reporting guidelines for Delphi? I cannot see this appended to the submission. <https://www.equator-network.org/reporting-guidelines/credes/> this would seem to be more important than SQUIRES since the methodology was explicitly framed as Delphi.

Can you explain or re-word in the paper what the phrase 'preliminary agreement between panel members' refers to? You state that 'initial rounds' involved open ended questions, and then later go into detail about the 4 rounds. This section therefore needs condensing and clarifying so there is not repetition. (same goes for participants where information is duplicated or introduced in two places).

Who were the participants? You contrast this with the next sentence which indicates that the initial rounds were a sub-group of the whole sample. Please provide more detail. This will mean some editing of the methods section (including e.g. p8-9) as well as the abstract.

Delbecq citation is not in the reference list. It is not a citation I know, but presumably some of the assertions about sample size rest on how heterogenous the sample is and how specific the research question is. How did your study fit with those assumptions?

Please explain what MOHW is.

Your sample description rests on some territory-specific assumptions. Can you clarify (for me and also in the paper) who

your sample was. In one sentence it indicates that all Delphi respondents were both practicing doctors and had a dual/second qualification in law? This is not a usual combination from where I live/work, but looks very interesting and helpful! But in a later sentence you say one was a social worker and one a clinical psychologist.

Why did you chose these specialities? Or was it an opportunity sample? Or is the GP/lawyer/social worker element tied to a different role of 'moderators' – in which case can you explain what a moderator is and how this is different to the sample. What is a moderator?

Remove reference to the in person conference if this was not focused on the Delphi training curricula.

I did not understand the 1st survey round. You indicate that the sample was asked about their background and then it becomes unclear – but appears that you asked to reflect on a specific fictionalised (?) conflict case. From that you distilled what frameworks they were implicitly drawing upon in their responses. You analysed whether they met core curricula of TIME (but in the introduction TIME is framed as a task-force not a training programme). It is unclear how this all then led to generating a questionnaire. How did this process of asking people to respond to a vignette add to a more standard Delphi approach of either qualitative interviews, or an analysis of the literature?

You repeat the threshold of consensus at 70%. It only needs stating once. Noting however that in the results and discussion sections, this is changed to 80%. Which is correct and why? What of dissensus?

Page 9 you use the word 'correlate' which I think needs to be removed, or the statistical understanding is likely to obfuscate how you want to communicate your process.

Were ethical approvals in place for this study?

Results

I did not understand the following phrase: "contentious proceeding in law case with medical errors: conventional means". Can you explain for the reader what this refers to.

The curricula appears to be about knowledge, rather than skills. Does the research literature indicate that it is a knowledge gap that causes or exacerbates medical disputes? Or is it skills? Only one small part of Table (conflict resolution) appears to be about actually addressing conflict. Had hoped to achieve consensus on what information people want in a curricula, rather than a skills-based curricula to enable change?

Table 4 introduces the idea of how much time each element takes. Where did this come from? I cannot see in the methods or results that respondents were asked about how to balance it or allocate training time.

Discussion: some of the information contained here would be better placed, in condensed form, the introduction, e.g. setting the context for the amount of medical disputes in Taiwan. However, I note that on page 12, there is a focus on skills which does not flow through to the curricula developed through your method. This disconnect warrants some discussion and exploration for the reader.

	On page 13, the text indicates that TIME training is already being used and evaluated. 'The ethics of TIME training encouraged inexperienced staff to exhibit empathy, keep an open mind, and be proficient in providing emotional support through situational exercise.' Is the reader to understand that following development of the curricula described in this paper, that you have then initiated its use and collected feedback? The section on looking after medical staff did not tie in closely with the narrative. Although I can see that disputes are difficult for staff, the section stands at odds to the rest of the paper. Similar point with the final section on 'continuity of TIME' which is not connected to the findings of the paper. This discussion section felt as though it was connected with a different/broader paper on TIME, rather than this specific one. I think it needs a lot of tightening up to make it specific to the findings of the Delphi study. References Reference 15: you have the author's first name, rather than last name in the list (should be Sumsion, T) 40 also includes a first name. Note that your citation of tables (references 19-21) will need formatting in house-style too.
--	--

VERSION 1 – AUTHOR RESPONSE

Reviewer: 1
Dr. Rano Mal Piryani, Universal College of Medical Sciences

Comments to the Author:

(1) It seems authors have contributed their maximum efforts in conducting this study and writing this manuscript.

Thank you for your encouragement and constructive comments about our manuscript.

Reviewer: 2
Jin-Ding Lin , Mackay Medical College

Comments to the Author:

(01) The necessity of this study need to emphasize to increase the importance in this study.

Thank you for the practical advice. The revision is shown as below:

With the evolution of legislation, the draft act for the prevention and resolution of medical malpractice was proposed by the Executive Yuan in 2018. Under the legal framework, active dispute care can alleviate the high tensions in doctor-patient relationships strained by medical disputes. A great demand for caring talents is anticipated, leading to the start of an associated programme named the triple-I mediator (TIME), authorized by the MOHW and characterized by 'in-house argument handling', 'immediate action', 'integration resources', to fix this talent gap. The

triple-I mediator (TIME) program is the first formal training programme, born in response to address prevention of dispute turning into litigation on the legal basis of Article 82 of the Medical Care Act, and borrowed from the concept of medical emergency team^{9, 10} or rapid response team¹¹ for the purpose of high acuity response to early deterioration of relationships between patients and physicians.

TIME, as the beginning of a five-year quality improvement series since 2018, a number of invited experts will attend a consensus meeting, conducted in a four-round Delphi process, for the extraction of experts' experiences as the base of constructing a syllabus, including teaching knowledge and skills in the mediation field, to educate newcomers to become competent mediators. The study aims to identify the core competencies and confirm the teaching materials for future tutors to provide precise instructions to trainees.

(02) The result of the abstract should have deep description instead of general information to the competence.

We appreciate your kind reminder of our oversight. We have revised the result of abstract as follows:

Results

Eleven enrolled experts reached the consensus about the training syllabus based on the 4-round agreement with four pillars of core competence, including 'knowledge base of law', 'internalisation of the denotative and connotative meanings of care', 'effective, smooth, and timely communication', and 'conflict resolution'. To grasp the dynamics and diversity of medical disputes on target, it is necessary to have sufficient knowledge and skills. We arrange our course in the order of teaching materials with pure didactics in the former two and with mixed contents comprising lectures and field exercises in the rest two.

(03) The author should have a brief description of previous training or teaching method, particularly in its pros and cons.

Thank you for your constructive critique. The articles in relation to training programs were conducted by researchers at hospital or institution level. The government-supported training program for cultivating relevant talents is scarcely published. The changes have been made per your recommendation:

Rapid reviews

'Rapid reviews' is a useful tool accepted by the World Health Organization (WHO) to furnish evidence in the target field in a cost-effective fashion¹⁷. References with respect to dispute resolving and relevant training programs, reviewed by the moderator, would be provided to the participants in the first Delphi round. The spectrum of keywords would be set up as wide as possible to include enough publications in the field of training courses for resolving medical disputes (i.e., 'medical dispute', 'conflict' and 'training'). Articles published between 1998 and 2018, written in English, and indexed in common databases, such as PubMed, Medline, and Embase, would be included. The sample considered whether they exhibited pillars of professionalism in the TIME programme based on relevant content in medical education¹⁸, Roach's Six C's in caring¹⁹, and a communication model called CI-CARE proposed by the University of California, Los Angeles

Health System²⁰, as a benchmark. Furthermore, evidence unveiled mediation training could facilitate healthcare staff to resolve conflicts at an early stage²¹. The basics of training should include several aspects of conflicts, comprising the identification of conflict resources, application of conflict management, and upgrading nonverbal communication skills²². A follow-up interval of six months after the training program showed that the trainees became more competent in coping with conflicts²³.

(04) The panel group was chosen of approved by the Taiwan's MOHW, is it have selection bias? it should have a discussion.

Following your recommendations, we have revised related sentences about purposive sampling and derivative limitations as follows:

The sample of experts in the TIME program

The experts participating in the TIME program were formally recruited in the manner of purposive sampling through granted consent of the expert invitation document sent by TDRF. These experts had more than 10 years of experience making sufficient contributions in the field of medical personnel-patient relationship repair, including serving as local civil mediators in various courts affiliated to the Judicial Yuan, and non-government organizations (NGOs) that provided legal services to patients/family members. Although the number of physicians outweighed other types of profession, it did not mean they stand the physicians' point of view. Instead, two of them long-term provided mental support and counselling to patients who were stuck in a poor relationship with their medical healthcare providers. In order to make sure a whole coverage of experts' opinions, the common triad encountered in medical disputes, including the arm of medical personnel, of patients/relatives, and the arm of legal professionals, were included in our study.

Limitations:

Some limitations must be noted. First, the sample composition was imbalanced due to an uneven distribution of specialties caused by the method of purposive sampling. Doubts arise as any possibility of selection bias still left. Second,

(05) The effect of the panel member occupation toward this study should have a discussion in the content.

The effect should be clarified to let our readers not to be confused about. We have emphasized what is new in our study and added information as follows:

The sample of experts in the TIME program

The experts participating in the TIME program were formally recruited in the manner of purposive sampling through granted consent of the expert invitation document sent by TDRF. These experts had more than 10 years of experience making sufficient contributions in the field of medical personnel-patient relationship repair, including serving as local civil mediators in various courts affiliated to the Judicial Yuan, and non-government organizations (NGOs) that provided legal services to patients/family members. Although the number of physicians outweighed other types of profession, it did not mean they stand the physicians' point of view. Instead, two of them long-term provided mental support and counselling to patients who were stuck in a poor relationship with their medical healthcare providers. In order to make sure a whole coverage of experts' opinions, the

common triad encountered in medical disputes, including the arm of medical personnel, of patients/relatives, and the arm of legal professionals, were included in our study.

Reviewer: 3

Prof. Liz Forbat, University of Stirling, Australian Catholic University - Canberra Campus

Comments to the Author:

I found this paper had promise and detailing the provenance of a training programme in medical conflict/dispute management has a lot to offer. There were a number of areas in the paper which require tightening up however, as detailed below.

(01) Title: add in Delphi method and the purpose of the paper, to indicate that this paper is about developing an educational intervention via a specific research method.

Thank you for pointing this out. We have changed the title shown as below:

The program of Triple-I mediator education (TIME) to improve medical disputes in clinical settings in Taiwan: a Delphi study

(02) Abstract: include sample size.

The sample size is added in our rephrased version of abstract shown as below:

Objectives

To establish a training programme to cultivate trainee mediation skills through time investment, skill incorporation, and formation of in-house mediation services.

Design

A four-round consensus conference was conducted by a number of seasoned experts selected in the manner of purposive sampling to determine core competences and relevant curricula through the modified Delphi process.

Setting

Responses collected from enrolled experts through four rounds of the Delphi process from 11th November 2018 to 17th May 2019.

Participants

Onboard seasoned mediators with different specialties.

Outcome measures

Items with a median rating of 4 or more on a Likert scale of 1 to 7 points and 70% or more in agreement were identified as core competence and curricula.

Results

Eleven enrolled experts reached the consensus about the training syllabus based on the 4-round agreement with four pillars of core competence, including 'knowledge base of law', 'internalisation of the denotative and connotative meanings of care', 'effective, smooth, and timely communication', and 'conflict resolution'. To grasp the dynamics and diversity of medical disputes on target, it is necessary to have sufficient knowledge and skills. We arrange our course in the order of teaching materials with pure didactics in the former two and with mixed contents comprising lectures and field exercises in the rest two.

(03) Conclusion – is it just between staff and patient? There is a large body of work on conflict in pediatric settings which is framed around the triad dynamic, i.e the position of a relative into the conflict, which is very common and can increase the potential for medical disputes when the patient cannot advocate for themselves.

The truth is the same as your description seen in our daily routine, that relatives are an inalienable part of the patient's arm. We have revised the paragraph based on your recommendation:

Conclusions

The sample developed a syllabus to train apprentices to take intermediate responses to medical disputes through the skills of conflict resolution and establishment of effective communication to improve the relationship between patients/relatives and medical staff, as a result of eventually reducing the conversion rate from dispute into litigation or alternative pathway. Policymakers in healthcare and top management in healthcare institutions can use this syllabus to guide their future education and training programme.

(04) Strengths: perhaps the language 'panel group' would be better replaced with the idea of 'sample'. Reconsider 'naive trainers' as this phrase isn't easily understandable without further context.

The section 'Strengths and limitations of this study' has been totally rephrased. The latest version is shown as below:

- ▶ The important strength of this study is the use of Delphi method providing anonymity to bring about detailed discussions.
- ▶ This study is the first official training program in Taiwan to train newcomers' skills in first-line mediation in clinical settings.
- ▶ This study is the beginning of a five-year Quality Improvement Program conducted by the Ministry of Healthcare and Welfare since 2018 and has the potential to be dedicated to policy improvement in fixing the broken and fragile relationships between medical personnel and patients/relatives.
- ▶ Limitations of our study include potentially selection bias caused by the method of purposive sampling and a lack of a follow-up system to assess training results.

(05) Introduction

(05-01) Can you define 'medical dispute'? is this different to conflict? Or litigation? Is it only between staff and patients, or intra-staff or intra-organisational conflict too?

Since TIME is defined as about dispute prevention, how does this sit alongside ideas of dispute management/resolution? They are on a continuum but if one is about prevention and one about resolution then they need to be conceptually presented clearly.

The concept of medical dispute should be clearly defined to make our readers to know the targets we focus on. Thank you for the practical advice. The revised one is shown as below:

The so-called medical dispute exists in a medical relationship but does not mean to enter into litigation. The concept of medical dispute tends to be an intermediate phase between the preliminary status, which we say 'conflict' as a disagreement occurs between medical staff and patients/relatives, and the final stage, including post-dispute problem-solving strategies with intra-judicial litigations and extra-judicial alternative dispute resolutions (ADRs).

(05-02) Can you provide more information on what TIME task force are 'seasoned experts' in – (A) what do they do, (B) how many are there, (C) what training have they had, (D) how are they selected, (E) what evaluations have been conducted? (F) You mention there's no systematic course 'in the long run' but it isn't clear what this means – is there a short course they do?

The seasoned experts should be depicted in detail for reviewers and readers to get a whole picture of who they are. The advice is concrete and practical. We will rephrase the relevant parts in our manuscript in line with this advice.

(A) These experts had more than 10 years of experience making sufficient contributions in the field of medical personnel-patient relationship repair, including serving as local civil mediators in various courts affiliated to the Judicial Yuan, and non-government organisations (NGOs) that provided legal services to patients/family members. (B) There are 11 experts enrolled in our study. (C) No universal training among these experts. These dedicated staff in charge of dispute management could only deal with their affairs by relying on individual professions, such as medical professions, legal professions, psychology professions, social work professions, and experience. (D) The experts participating in the TIME program were formally recruited in the manner of purposive sampling through granted consent of the expert invitation document sent by TDRF. (E) The evaluations have been conducted based on their own contributions to relevant fields in medical dispute mediation to decide whether staff could be enrolled in this study. (F) Until 2018, there was no officially educational program of medical dispute resolution training in Taiwan.

(05-03) I was expecting to see a brief section detailing what other trainings are available for managing disputes in medical settings, internationally. One such approach has not been cited, but there may be others. <https://pubmed.ncbi.nlm.nih.gov/27098546/> This is especially important given that this was part of your methods process as stated on p7.

The brief section is added based in the 'Method' on your recommendations:

Rapid reviews

'Rapid reviews' is a useful tool accepted by the World Health Organization (WHO) to furnish evidence in the target field in a cost-effective fashion¹⁷. References with respect to dispute resolving and relevant training programs, reviewed by the moderator, would be provided to the

participants in the first Delphi round. The spectrum of keywords would be set up as wide as possible to include enough publications in the field of training courses for resolving medical disputes (i.e., 'medical dispute', 'conflict' and 'training'). Articles published between 1998 and 2018, written in English, and indexed in common databases, such as PubMed, Medline, and Embase, would be included. The sample considered whether they exhibited pillars of professionalism in the TIME programme based on relevant content in medical education¹⁸, Roach's Six C's in caring¹⁹, and a communication model called CI-CARE proposed by the University of California, Los Angeles Health System²⁰, as a benchmark. Furthermore, evidence unveiled mediation training could facilitate healthcare staff to resolve conflicts at an early stage²¹. The basics of training should include several aspects of conflicts, comprising the identification of conflict resources, application of conflict management, and upgrading nonverbal communication skills²². A follow-up interval of six months after the training program showed that the trainees became more competent in coping with conflicts²³.

(06) Methods

(06-01) Can you explain why patients/public were not involved? Since medical disputes are framed in the introduction as between health care staff and patients (i.e., not between healthcare staff) then it might have been useful to integrate their perspectives.

Thank you for the constructive comments. The explanation about why the public/patients are not involved in the study is shown as below:

This TIME program is an educational project focusing on the handling of disputes between medical staff and patients/relatives at the beginning of a dispute in order to achieve the goal of reducing the rate of disputes being transformed into litigation/ alternative dispute resolution (ADR). The educational materials would be under construction based on seasoned expertise's perspectives on coping strategies in this field. The purpose of invitations of experts in relevant fields is to extract expert experience and convert it into the basis of a systematic curriculum. To balance the weight of the opinions of medical staff and patients/relatives, different voices and opinions should come from both arms rather than one side. Our strategy was to use purposive sampling to appropriately recruit the equivalents of the doctor (such as clinicians) and the equivalents of the patient (such as legal experts responsible for handling medical disputes).

(06-02) Did you use the CREDES reporting guidelines for Delphi? I cannot see this appended to the submission. <https://www.equator-network.org/reporting-guidelines/credes/> this would seem to be more important than SQUIRES since the methodology was explicitly framed as Delphi.

Thank you for pointing out this problem. The CREDES checklist would be added this time. The revision is shown as follows:

The Delphi method

The Delphi method is reliable and has been effectively used for establishing consensus regarding healthcare management¹³. It is a multistage, iterative, and anonymous process consisting of at least two rounds of surveys¹⁴. Opinions and feedback were structurally categorized under different headings after each round and exported to a temporary meeting document as reference to effectively facilitate discussions among moderators in the next round. Considering the critical paucity of relevant evidence for fostering skills, the Delphi method with four rounds of surveys was introduced to achieve consensus regarding the syllabus for the TIME programme. The strengths of this method included anonymity, capability of obtaining a collection of ideas and opinions from participants, and cost effectiveness. The weaknesses were the length of time required, research

bias through the potential influence of moderators or participants, and uneven distribution of specialties among the respondents. The authors followed the guidance of Conducting and REporting DElphi Studies (CREDES)¹⁵.

(06-03) Can you explain or re-word in the paper what the phrase 'preliminary agreement between panel members' refers to?

The preliminary agreement is to describe the convergence of the points of view about the reviewed articles among participants. However, this description confused reviewers/future readers and would be revised as below:

After a review of research studies, the initial round with open-ended questions was initiated to elicit various thoughts and perspectives.

(06-04) You state that 'initial rounds' involved open ended questions, and then later go into detail about the 4 rounds. This section therefore needs condensing and clarifying so there is not repetition. (same goes for participants where information is duplicated or introduced in two places).

Initial rounds would be re-worded to become a new part named 'Rapid reviews', and the revision is shown as below:

Rapid reviews

'Rapid reviews' is a useful tool accepted by the World Health Organization (WHO) to furnish evidence in the target field in a cost-effective fashion¹⁷. References with respect to dispute resolving and relevant training programs, reviewed by the moderator, would be provided to the participants in the first Delphi round. The spectrum of keywords would be set up as wide as possible to include enough publications in the field of training courses for resolving medical disputes (i.e., 'medical dispute', 'conflict' and 'training'). Articles published between 1998 and 2018, written in English, and indexed in common databases, such as PubMed, Medline, and Embase, would be included. The sample considered whether they exhibited pillars of professionalism in the TIME programme based on relevant content in medical education¹⁸, Roach's Six C's in caring¹⁹, and a communication model called CI-CARE proposed by the University of California, Los Angeles Health System²⁰, as a benchmark. Furthermore, evidence unveiled mediation training could facilitate healthcare staff to resolve conflicts at an early stage²¹. The basics of training should include several aspects of conflicts, comprising the identification of conflict resources, application of conflict management, and upgrading nonverbal communication skills²². A follow-up interval of six months after the training program showed that the trainees became more competent in coping with conflicts²³.

Delphi survey rounds

In round 1, the sample was required to read aforementioned references reviewed by the moderator and provided details of their occupations, experience with regard to dealing with medical disputes in clinical settings, knowledge of conflict management, and coping strategies used to alleviate the high tensions between medical personnel and patients/relatives. Opinions and details on constructing core competences correlated with fostering professionalism and reflection after reading references were provided using narrative text responses and discussion. The feedback and views of experts were carefully gathered to generate a draft questionnaire.

In round 2, the draft proposal comprising several features of core competences was presented and measured through a questionnaire answered using a 7-point Likert-type scale (1 = *strongly disagree*, 2 = *disagree*, 3 = *somewhat disagree*, 4 = *neutral*, 5 = *somewhat agree*, 6 = *agree*, and 7 = *strongly agree*). Consensus was based on an agreement level of >70%. Items with <70% agreement and relevant feedback regarding disagreement with a core competence was discussed by the moderator group. Participants were then requested to share opinions and details for constructing the core curricula correlating to fostering core competences by using narrative text responses. Information on positive and negative aspects mentioned earlier were gathered systematically to generate the initial draft containing the core competences and core curricula of the TIME programme.

In round 3, agreement among the panel members regarding the draft of the core curricula was obtained using a 7-point Likert scale similar to that used in round 2 with respect to the level of agreement by the respondents. Consensus was reached and discussions were held by the moderator group if any item had <70% agreement. Additional narrative text was provided to determine whether any disagreement existed regarding the core curriculum. This draft was revised by the group based on the analysis of agreement and annotation of round 3. Besides, participants were asked to calculate the time in each segment of this training program in self-simulated situations and the scheduled class time would be discussed in the fourth round.

In round 4, respondents were asked to provide suggestions on syntactic errors, improper phrasing, or terminology to make the description concise. The final version of the TIME training syllabus included (1) immediate action (2) policy incorporation (3) in-house settlement based on agreement in rounds 2 and 3. In regard to the class time of each segment, including 'Knowledge base of law', 'Internalisation of the denotative and connotative meanings of care', 'Effective, smooth, and timely communication', 'Conflict resolution', would be one, one, three, and three hours, respectively, after the participants reached the consensus after taking the reasonable length of content in each segment of this course into accounts. The final version of the syllabus draft was validated and announced on 15 November 2019, the promulgation date (Figure 1), accompanied with the announcement of the initial version of the official course handbook based on the draft by the MOHW. The latest version on the official website of the MOHW was free to the public with open access²⁴.

(06-05) Who were the participants? You contrast this with the next sentence which indicates that the initial rounds were a sub-group of the whole sample. Please provide more detail. This will mean some editing of the methods section (including e.g. p8-9) as well as the abstract.

The participants are those experts in the field of fixing personnel-patient relationships. They thoroughly take part in four Delphi rounds from the beginning to the end of the study. The details with respect to experts is added in the revision:

(01) The sample of experts in the TIME program

The experts participating in the TIME program were formally recruited in the manner of purposive sampling through granted consent of the expert invitation document sent by TDRF. These experts had more than 10 years of experience making sufficient contributions in the field of medical personnel-patient relationship repair, including serving as local civil mediators in various courts affiliated to the Judicial Yuan, and non-government organizations (NGOs) that provided legal services to patients/family members. Although the number of physicians outweighed other types of profession, it did not mean they stand the physicians' point of view. Instead, two of them long-term provided mental support and counselling to patients who were stuck in a poor relationship with their medical healthcare providers. In order to make sure a whole coverage of experts' opinions, the

common triad encountered in medical disputes, including the arm of medical personnel, of patients/relatives, and the arm of legal professionals, were included in our study.

(02) Abstract

Objectives

To establish a training programme to cultivate trainee mediation skills through time investment, skill incorporation, and formation of in-house mediation services.

Design

A four-round consensus conference was conducted by a number of seasoned experts selected in the manner of purposive sampling to determine core competences and relevant curricula through the modified Delphi process.

Setting

Responses collected from enrolled experts through four rounds of the Delphi process from 11th November 2018 to 17th May 2019.

Participants

Onboard seasoned mediators with different specialties.

Outcome measures

Items with a median rating of 4 or more on a Likert scale of 1 to 7 points and 70% or more in agreement were identified as core competence and curricula.

Results

Eleven enrolled experts reached the consensus about the training syllabus based on the 4-round agreement with four pillars of core competence, including 'knowledge base of law', 'internalisation of the denotative and connotative meanings of care', 'effective, smooth, and timely communication', and 'conflict resolution'. To grasp the dynamics and diversity of medical disputes on target, it is necessary to have sufficient knowledge and skills. We arrange our course in the order of teaching materials with pure didactics in the former two and with mixed contents comprising lectures and field exercises in the rest two.

Conclusions

The sample developed a syllabus to train apprentices to take intermediate responses to medical disputes through the skills of conflict resolution and establishment of effective communication to improve the relationship between patients/relatives and medical staff, as a result of eventually reducing the conversion rate from dispute into litigation or alternative pathway. Policymakers in healthcare and top management in healthcare institutions can use this syllabus to guide their future education and training programme.

(06-06) (A) Delbecq citation is not in the reference list. It is not a citation I know, but presumably some of the assertions about sample size rest on how heterogenous the sample is and how specific the research question is. (B) How did your study fit with those assumptions?

The citation and responses to the question of assumptions have been rephrased as follows:

(A) The correct citation is 'Andre L Delbecq, Andrew H. Van de Ven and David Gustafson, *Group Techniques for Program Planning; a guide to nominal group and Delphi processes*. Glenview IL, Scott Foresman and Company, 1975. ISBN 0-673-07591-5'.

(B) The experts participating in the TIME program were formally recruited in the manner of purposive sampling through granted consent of the expert invitation document sent by TDRF. These experts had more than 10 years of experience making sufficient contributions in the field of medical personnel-patient relationship repair, including serving as local civil mediators in various courts affiliated to the Judicial Yuan, and non-government organizations (NGOs) that provided legal services to patients/family members. Although the number of physicians outweighed other types of profession, it did not mean they stand the physicians' point of view. Instead, two of them long-term provided mental support and counselling to patients who were stuck in a poor relationship with their medical healthcare providers. In order to make sure a whole coverage of experts' opinions, the common triad encountered in medical disputes, including the arm of medical personnel, of patients/relatives, and the arm of legal professionals, were included in our study.

(06-07) Please explain what MOHW is.

The Ministry of Health and Welfare (MOHW) is with affiliation to the Executive Yuan ministry, one of five pillars in taking responsibility to main normal functioning of country, including Legislative Yuan, Judicial Yuan, Control Yuan, and Examination Yuan. The MOHW is in charge of the administration of the public health system, the cooperation of social welfare network and universal healthcare service and the mission of MOHW is to promote the health and well-being of all citizens by operating programs with regards to health policy and management, including healthcare institutions, pharmaceutical, immunization programs, disease prevention, supervision, and coordination of local health agencies in Taiwan.

(06-08) Your sample description rests on some territory-specific assumptions. Can you clarify (for me and also in the paper) who your sample was. In one sentence it indicates that all Delphi respondents were both practicing doctors and had a dual/second qualification in law? This is not a usual combination from where I live/work, but looks very interesting and helpful! But in a later sentence you say one was a social worker and one a clinical psychologist.

Thank you for the critical comments. My previous version seems to make readers confused. There are six physicians, two lawyer, one judge, one clinical psychologist, and social worker enrolled in our study. The rephrased version is shown as below:

The sample of experts in the TIME program

The experts participating in the TIME program were formally recruited in the manner of purposive sampling through granted consent of the expert invitation document sent by TDRF. These experts had more than 10 years of experience making sufficient contributions in the field of medical personnel-patient relationship repair, including serving as local civil mediators in various courts affiliated to the Judicial Yuan, and non-government organizations (NGOs) that provided legal services to patients/family members. Although the number of physicians outweighed other types of profession, it did not mean they stand the physicians' point of view. Instead, two of them long-term provided mental support and counselling to patients who were stuck in a poor relationship with their medical healthcare providers. In order to make sure a whole coverage of experts' opinions, the

common triad encountered in medical disputes, including the arm of medical personnel, of patients/relatives, and the arm of legal professionals, were included in our study.

(06-09) (A) Why did you chose these specialities? Or was it an opportunity sample? Or is the GP/lawyer/social worker element tied to a different role of 'moderators' – in which case can you explain what a moderator is and how this is different to the sample. (B) What is a moderator?

(A)The precise description provided by the reviewer make the paragraph become better readable. This is literally purposive sampling. The experts participating in the TIME program were formally recruited in the manner of purposive sampling through granted consent of the expert invitation document sent by TDRF. (B) To integrate experts' opinions effectively, a clinical physician practicing family medicine with position as an associate professor in medicine and a Doctor of Laws degree, is designated as the moderator by the Taiwan Drug Relief Foundation (TDRF) due to his multiple experiences for interdisciplinary coordination, such as conducting a pilot project to encourage medical institutions to properly handle surgical and anaesthesia disputes in 2014 and bringing about the satisfactory outcome of the promotion of the Childbirth Accident Emergency Relief Act in 2015.

(06-10) Remove reference to the in person conference if this was not focused on the Delphi training curricula.

This has been revised and shown as below:

The MOHW supported this TIME project and conducted a 4-round, in-person, consensus conference on November 11, 2018, January 1, March 11, and May 17, 2019, separately, to reach the consensus about the syllabus for the course of TIME training through the Delphi method as the beginning of a five-year Quality Improvement Program.

(06-11) I did not understand the 1st survey round. You indicate that the sample was asked about their background and then it becomes unclear – but appears that you asked to reflect on a specific fictionalised (?) conflict case. From that you distilled what frameworks they were implicitly drawing upon in their responses. You analysed whether they met core curricula of TIME (but in the introduction TIME is framed as a task-force not a training programme). It is unclear how this all then led to generating a questionnaire. How did this process of asking people to respond to a vignette add to a more standard Delphi approach of either qualitative interviews, or an analysis of the literature?

References with respect to dispute resolving and relevant training programs, reviewed by the moderator, would be provided to the participants in the first Delphi round. We have written a paragraph about the relevant literature review in topics of dispute resolving and training programs in other countries in the section 'Method'. The context of the first Delphi round and subsequent sentences is not clear in the current expression, and it would be modified to reach a better level of understanding. The revised version is shown as below:

Rapid reviews

'Rapid reviews' is a useful tool accepted by the World Health Organization (WHO) to furnish evidence in the target field in a cost-effective fashion¹⁷. References with respect to dispute resolving and relevant training programs, reviewed by the moderator, would be provided to the participants in the first Delphi round. The spectrum of keywords would be set up as wide as

possible to include enough publications in the field of training courses for resolving medical disputes (i.e., 'medical dispute', 'conflict' and 'training'). Articles published between 1998 and 2018, written in English, and indexed in common databases, such as PubMed, Medline, and Embase, would be included. The sample considered whether they exhibited pillars of professionalism in the TIME programme based on relevant content in medical education¹⁸, Roach's Six C's in caring¹⁹, and a communication model called CI-CARE proposed by the University of California, Los Angeles Health System²⁰, as a benchmark. Furthermore, evidence unveiled mediation training could facilitate healthcare staff to resolve conflicts at an early stage²¹. The basics of training should include several aspects of conflicts, comprising the identification of conflict resources, application of conflict management, and upgrading nonverbal communication skills²². A follow-up interval of six months after the training program showed that the trainees became more competent in coping with conflicts²³.

Round 1

In round 1, the sample was required to read aforementioned references reviewed by the moderator and provided details of their occupations, experience with regard to dealing with medical disputes in clinical settings, knowledge of conflict management, and coping strategies used to alleviate the high tensions between medical personnel and patients/relatives. Opinions and details on constructing core competences correlated with fostering professionalism and reflection after reading references were provided using narrative text responses and discussion. The feedback and views of experts were carefully gathered to generate a draft questionnaire.

(06-12) You repeat the threshold of consensus at 70%. It only needs stating once. Noting however that in the results and discussion sections, this is changed to 80%. Which is correct and why? What of dissensus?

The threshold of consensus is defined as one above the level of neutral and comprised 70% of population. The threshold of consensus is 70%. The agreement more than 80% is just the conclusive description of agreement level in individual items, such as in 80.2% 'Knowledge base of law', 100% in 'Internalisation of the denotative and connotative meanings of care', 100% in 'Effective, smooth, and timely communication', and 100% in 'Conflict resolution'. However, this kind of description could confuse both the reviewers and readers. The revised version would be shown as below:

In the section 'Results':

Table 2 shows a consensus on all items, with more than 70% agreement. The item 'effective, smooth, and timely communication' had the highest number of responses of 'strongly agree' (n = 10, 90.9%), and the rest of the items, including 'knowledge base of law', 'internalisation of the denotative and connotative meanings of care', and 'conflict resolution', had almost equal agreement (n = 6, 54.5%). Only the item 'knowledge base of the law' elicited a neutral opinion (n = 2, 18.2%).

In the section 'Discussion':

After a four-round Delphi process was applied, consensus was reached regarding the syllabus for the TIME training programme, which was established by a panel consisting of highly experienced experts in mediation. Table 4 was the final product containing four items on core competencies with respect to relevant core curricula. Each item received at least 70% agreement, which represents convergence of opinions.

(06-13) Page 9 you use the word 'correlate' which I think needs to be removed, or the statistical understanding is likely to obfuscate how you want to communicate your process.

We have revised the sentences based on your recommendations:

In table 3, a consensus was reached for all items, with an agreement of 100%, except for the item 'contentious proceeding in law case with medical errors: conventional means', which had 91.7% agreement. Only in the 'knowledge base of law' category was the number of participants who responded of strongly agree below 10 (n = 9). The other three categories had at least one item with >10 people indicating strong agreement. Additionally, in the categories of 'effective, smooth, and timely communication' and 'conflict resolution', consensus was reached on all of the ~~correlated~~ core curricula, with more than half of the members in strong agreement. Furthermore, only the category of 'effective, smooth, and timely communication' had two items for which 10 members were in strong agreement, namely 'ethics of TIME: knowing the difference between what you have a right to do and what is right to do' and 'Establishment of communication: an impregnable foundation leaving no boundary and obstacle.

(06-14) Were ethical approvals in place for this study?

The added information have been included in our manuscript:

Ethics approval and consent to participate

The TIME program was approved by the Institutional Review Board (IRB) and the Research Ethics Committee of the Tri-Service General Hospital (TSGHIRB: E202216022) and was conducted in line with the Declaration of Helsinki. Informed consent was obtained along with the participants' responses.

(07) Results

(07-01) I did not understand the following phrase: "contentious proceeding in law case with medical errors: conventional means". Can you explain for the reader what this refers to.

This could confuse the reviewers and the readers. The use of 'contentious' would be re-worded to make it a better level of understanding. The revision is shown as below:

'Proceeding in law case with medical disputes: conventional means' means that the dispute would be resolved by the institution of legal action rather than the alternative dispute resolution.

(07-02) The curricula appears to be about knowledge, rather than skills. Does the research literature indicate that it is a knowledge gap that causes or exacerbates medical disputes? Or is it skills? Only one small part of Table (conflict resolution) appears to be about actually addressing conflict. Had hoped to achieve consensus on what information people want in a curricula, rather than a skills-based curricula to enable change?

In our expert consensus, mediators should strengthen the training of communication skills, including the establishment of effective communication and dispute resolution. Therefore, in the course, we arrange the first hour of knowledge courses, followed by two hours of field exercises. The aforementioned information is added in the revised manuscript:

(1) Strengthening the capability of smooth and effect communication:

The core concept conveyed in the segment 'Effective, smooth, and timely communication' is that the quality of care would be maintained by the high-quality of patient-centred caring (PCC)^{32, 33}. As the first step of PCC, demonstrating empathy, including verbal and non-verbal (e.g., body expressions³⁴) clues in every visit, is a critical and teachable multi-phrase skill³⁵. Six keys are taught in our course to facilitate effective empathy³⁶, including (1) discerning existence of emotions in the clinical setting, (2) taking a pause to imagine what the interviewer (e.g., medical personnel or patients/relatives) may think about, (3) stating mediators' perception of the interviewer's feeling (i.e., "It sounds like this event really frustrates you in depth ..."), (4) making the stated feeling presented by the interviewer plausible, (5) showing respect for the interviewer's effort to tackle the dilemma, and (6) furnishing mental support and partnership (i.e., "Let's move to the action that we can take together to..."). The trainers would teach trainees a formula comprising queries-clarifications- in order to memorize and to practice³⁷.

Trainees would learn the ropes of integrating emotional clues about physical or mental health from tutors in our teaching program. These collected threads will be the base for approaching the interviewee's involved in medical dispute authentic feelings. To be more specific, detected emotions could be approximately categorized in the Kübler-Ross Change Curve (KRCC), including grief denial, anger, bargaining, depression, acceptance, and coping strategies could be conducted according to individual differences. However, one thing should be kept in mind that this category is just an auxiliary measure for approaching the interviewer's authentic feelings. Stiffly putting someone's fragment of emotional journey into the framework of the KRCC is inappropriate³⁸. These detections of emotions by a well-educated mediator facilitate precise and rapid responses to such feelings because physical discomfort leads to mental and emotional suffering³⁹. As physical health decline, either temporarily or permanently, would have mutual⁴⁰ and detrimental effects on psychological health (e.g., anxiety⁴¹, depression⁴²), leading to eventually undermining the existing/established-in-the-process partnerships and difficulty in building trust with others⁴³. The two-hour practice in this segment of the TIME training program is designed to allow trainees to be proficient in providing emotional support through situational exercise and to receive verbal feedback on their performances by trainers.

(2) Strengthening the capability of smooth and effect communication:

When a medical dispute occurs, both medical personnel⁴⁴ and patients/relatives⁴⁵ would be influenced and become vulnerable, sensitive, and self-absorbed. The defense mechanism will be activated in both parties and cause the relationship to become increasingly alienated and destructive, and make people more injured, bringing about a form of vicious cycle. To break this cycle and restore the broken trust, the so-called 'transformative mediation⁴⁶' with two key features is fundamental and would be practiced in the practicing part of the course section 'Conflict resolution'. One is empowerment shift⁴⁷, which could allow both parties to regain respect, confidence, and empathy. The other is recognition shift⁴⁷, which could help patients reduce their defensiveness against medical staff and gain a different perspective. This approach made patients express needs and strengthened bilateral participation in health care issue⁴⁸.

(07-03) Table 4 introduces the idea of how much time each element takes. Where did this come from? I cannot see in the methods or results that respondents were asked about how to balance it or allocate training time.

We appreciate your kind reminder. In round 3, the apparent contour and content of the TIME program had almost been depicted except for the time each segment took. To make it perfect in the final round, participants were asked to calculate the time in each segment in self-simulated

situations and the scheduled class time would be discussed in the fourth round. The revision is shown as below:

In round 3, agreement among the panel members regarding the draft of the core curricula was obtained using a 7-point Likert scale similar to that used in round 2 with respect to the level of agreement by the respondents. Consensus was reached and discussions were held by the moderator group if any item had <70% agreement. Additional narrative text was provided to determine whether any disagreement existed regarding the core curriculum. This draft was revised by the group based on the analysis of agreement and annotation of round 3. Besides, participants were asked to calculate the time in each segment of this training program in self-simulated situations and the scheduled class time would be discussed in the fourth round.

In round 4, respondents were asked to provide suggestions on syntactic errors, improper phrasing, or terminology to make the description concise. The final version of the TIME training syllabus included (1) immediate action (2) policy incorporation (3) in-house settlement based on agreement in rounds 2 and 3. In regard to the class time of each segment, including 'Knowledge base of law', 'Internalisation of the denotative and connotative meanings of care', 'Effective, smooth, and timely communication', 'Conflict resolution', would be one, one, three, and three hours, respectively, after the participants reached the consensus after taking the reasonable length of content in each segment of this course into accounts. The final version of the syllabus draft was validated and announced on 15 November 2019, the promulgation date (Figure 1), accompanied with the announcement of the initial version of the official course handbook based on the draft by the MOHW. The latest version on the official website of the MOHW was free to the public with open access²⁴.

(08) Discussion

(08-01) (A) Some of the information contained here would be better placed, in condensed form, the introduction, e.g. setting the context for the amount of medical disputes in Taiwan. (B) However, I note that on page 12, there is a focus on skills which does not flow through to the curricula developed through your method. This disconnect warrants some discussion and exploration for the reader.

(A) The initial version of the paragraph 'Increases in medical disputes and civil and criminal proceedings: the current situation in Taiwan' in the section 'Discussion' is merged with the first paragraph of the section 'Introduction'. The revision is shown as below:

The number of medical disputes has been increasing¹. An intermediate phase is the meaning of medical dispute between the preliminary status, which 'conflict' goes as a disagreement between medical staff and patients/relatives, and the dispute resolution status as the final stage. Most of these disputes belong to non-negligence after factual approach of causation in retrospective analysis all the time². To resolve these disputes efficiently, several approaches have been utilized, including post-dispute problem-solving strategies with intra-judicial litigations and extra-judicial alternative dispute resolutions (ADRs)³⁻⁵. ADR has been integrated into the healthcare system of many countries⁶⁻⁸ for mitigating risk of being sued and making both patients and physicians satisfied^{3, 4}. Pre-dispute risk identification and early intervention, however, was merely merged into our current system.

(B) The revision is shown as below:

The quality of communication can determine the direction of the relationships between medical staff-patients/relatives. Poor communication stems from the gap in mutual understanding, moving to incredulity²⁸, and causing the overuse of defensive medicine as a non-violent strategy in the management of medical disputes²⁹. This psychological chain of reaction could be fixed from the

technical perspective of resolution management, such as using Google Translate, a famous applied science-oriented technology, helping overcome language barriers and improve satisfaction levels among healthcare providers and patients³⁰. The design of ideal mediators after our TIME training is not focused on technical aspects mentioned before but on in-person communication by arriving at the scene of the dispute immediately and providing immediate care. In hospital settings, a mediator can activate the integration of resources after engaging with both parties of a dispute. In clinical settings, patients focus on discomfort and ignore or miss relevant information regarding available treatments. The utility of this related mediation may be time sensitive. Mediators accelerate the use of individualised services targeting specific demands after the overall evaluation of case needs and concerns. For those with legal liability for any damage they cause, legal remedies affirm the validity of their claims and requests. Damage caused by those with no liability could be dealt with through compensation³¹. Common access points to humanitarian relief include Taiwan's drug injury relief system, compensation for vaccine injuries, and childbirth accident relief.

(08-02) On page 13, the text indicates that TIME training is already being used and evaluated. 'The ethics of TIME training encouraged inexperienced staff to exhibit empathy, keep an open mind, and be proficient in providing emotional support through situational exercise.' Is the reader to understand that following development of the curricula described in this paper, that you have then initiated its use and collected feedback?

Thank you for the important recommendations. The feedback collecting system is still under construction. The current feedback is provided orally and not collected in a systematic approach. This have been rephrased in our manuscript:

Trainees would learn the ropes of integrating emotional clues about physical or mental health from tutors in our teaching program. These collected threads will be the base for approaching the interviewee's involved in medical dispute authentic feelings. To be more specific, detected emotions could be approximately categorized in the Kübler-Ross Change Curve (KRCC), including grief denial, anger, bargaining, depression, acceptance, and coping strategies could be conducted according to individual differences. However, one thing should be kept in mind that this category is just an auxiliary measure for approaching the interviewer's authentic feelings. Stiffly putting someone's fragment of emotional journey into the framework of the KRCC is inappropriate³⁸. These detections of emotions by a well-educated mediator facilitate precise and rapid responses to such feelings because physical discomfort leads to mental and emotional suffering³⁹. As physical health decline, either temporarily or permanently, would have mutual⁴⁰ and detrimental effects on psychological health (e.g., anxiety⁴¹, depression⁴²), leading to eventually undermining the existing/established-in-the-process partnerships and difficulty in building trust with others⁴³. The two-hour practice in this segment of the TIME training program is designed to allow trainees to be proficient in providing emotional support through situational exercise and to receive verbal feedback on their performances by trainers.

(08-03) The section on looking after medical staff did not tie in closely with the narrative. Although I can see that disputes are difficult for staff, the section stands at odds to the rest of the paper. Similar point with the final section on 'continuity of TIME' which is not connected to the findings of the paper.

The section 'Looking after medical staff' has been emerged to the seventh paragraph in the section 'Discussion' shown as follows:

As we focused on the arm of patients/relatives, there would be ignorance of the need for care from the arm of medical personnel. This constantly neglected blind spot of calling for help for

healthcare professions is going to be smoothed by mediators after the TIME training. The potential emotional trauma of staff in response to adverse patient responses can cause burnout⁴⁹. Such burnout has been thought to be high among physicians globally⁵⁰. Consequences of burnout have detrimental effects on doctors, which may lead to them making incorrect decisions (thus causing medical errors), exhibiting a hostile attitude towards patients, and having dysfunctional relationships with colleagues⁴⁹. Once staff having traumatic experiences they would suffer from second victim syndrome (SVS)⁵¹. Mediators after the TIME training could recognize the six phrases, including the early stage of 'initial chaos and accident response' and the final of 'moving on'⁵². In half of the process during the SVS recovery trajectory, especially in phrase three, the suffering personnel would try to seek help and re-establish trust with surroundings⁵². This would be the timing on target for mediators to approach involved personnel by listening empathetically, eschew judgement, and showing respect⁵³. SVS cannot be treated if systematic and well-structured support for coping with emotional burdens is absent⁵⁴. To put it in a nutshell, medical staff should be provided with adequate care similar to support for patients/relatives with mature, intact, and positive counselling⁵⁵.

The section 'continuity of TIME' has been emerged to the second paragraph in the section 'Discussion' shown as follows:

To optimise the present healthcare system, Taiwan's MOHW introduced an in-house mediation (IHM) system for conflict management in 2013²⁵. According to current regulations on IHM in Taiwan, mediators are defined as sentinel and people with an engaging personality²⁶ who provide mediation for a suspected, gradually formed, or established dispute, and collaborate with team members from different specialties depending on the scale or type of dispute. The mediator is similar with a symphony conductor, setting the tempo and guaranteeing proper entries for the ensemble cast²⁷, to put fragment of narratives of the conflict together into euphony instead of cacophony. Of the total number of members in the team, at least three must be in-house medical staff in addition to the original teams of healthcare providers and nonmedical staff (e.g., legal professionals or social workers) and nonmedical staff should comprise at least one third of the team in line with law. The design of IHM is aimed at creating a 360° feedback mechanism and obtaining good understanding of excellence; in reality, the ultimate goal of IHM application in dispute resolution is to ensure highly effective teamwork for appropriate dispute resolution rather than blame shifting.

(08-04) This discussion section felt as though it was connected with a different/broader paper on TIME, rather than this specific one. I think it needs a lot of tightening up to make it specific to the findings of the Delphi study.

The section 'Discussion' has been rephrased largely to make the content more fluent and logic in a strict sense. The revision is shown as follows:

Discussion

After a four-round Delphi process was applied, consensus was reached regarding the syllabus for the TIME training programme, which was established by a panel consisting of highly experienced experts in mediation. Table 4 was the final product containing four items on core competencies with respect to relevant core curricula. Each item received at least 70% agreement, which represents convergence of opinions.

To optimise the present healthcare system, Taiwan's MOHW introduced an in-house mediation (IHM) system for conflict management in 2013²⁵. According to current regulations on IHM in

Taiwan, mediators are defined as sentinel and people with an engaging personality²⁶ who provide mediation for a suspected, gradually formed, or established dispute, and collaborate with team members from different specialties depending on the scale or type of dispute. The mediator is similar with a symphony conductor, setting the tempo and guaranteeing proper entries for the ensemble cast²⁷, to put fragment of narratives of the conflict together into euphony instead of cacophony. Of the total number of members in the team, at least three must be in-house medical staff in addition to the original teams of healthcare providers and nonmedical staff (e.g., legal professionals or social workers) and nonmedical staff should comprise at least one third of the team in line with law. The design of IHM is aimed at creating a 360° feedback mechanism and obtaining good understanding of excellence; in reality, the ultimate goal of IHM application in dispute resolution is to ensure highly effective teamwork for appropriate dispute resolution rather than blame shifting.

The quality of communication can determine the direction of the relationships between medical staff-patients/relatives. Poor communication stems from the gap in mutual understanding, moving to incredulity²⁸, and causing the overuse of defensive medicine as a non-violent strategy in the management of medical disputes²⁹. This psychological chain of reaction could be fixed from the technical perspective of resolution management, such as using Google Translate, a famous applied science-oriented technology, helping overcome language barriers and improve satisfaction levels among healthcare providers and patients³⁰. The design of ideal mediators after our TIME training is not focused on technical aspects mentioned before but on in-person communication by arriving at the scene of the dispute immediately and providing immediate care. In hospital settings, a mediator can activate the integration of resources after engaging with both parties of a dispute. In clinical settings, patients focus on discomfort and ignore or miss relevant information regarding available treatments. The utility of this related mediation may be time sensitive. Mediators accelerate the use of individualised services targeting specific demands after the overall evaluation of case needs and concerns. For those with legal liability for any damage they cause, legal remedies affirm the validity of their claims and requests. Damage caused by those with no liability could be dealt with through compensation³¹. Common access points to humanitarian relief include Taiwan's drug injury relief system, compensation for vaccine injuries, and childbirth accident relief.

The core concept conveyed in the segment 'Effective, smooth, and timely communication' is that the quality of care would be maintained by the high-quality of patient-centred caring (PCC)^{32, 33}. As the first step of PCC, demonstrating empathy, including verbal and non-verbal (e.g., body expressions³⁴) clues in every visit, is a critical and teachable multi-phrase skill³⁵. Six keys are taught in our course to facilitate effective empathy³⁶, including (1) discerning existence of emotions in the clinical setting, (2) taking a pause to imagine what the interviewer (e.g., medical personnel or patients/relatives) may think about, (3) stating mediators' perception of the interviewer's feeling (i.e., "It sounds like this event really frustrates you in depth ..."), (4) making the stated feeling presented by the interviewer plausible, (5) showing respect for the interviewer's effort to tackle the dilemma, and (6) furnishing mental support and partnership (i.e., "Let's move to the action that we can take together to..."). The trainers would teach trainees a formula comprising queries-clarifications- in order to memorize and to practice³⁷.

Trainees would learn the ropes of integrating emotional clues about physical or mental health from tutors in our teaching program. These collected threads will be the base for approaching the interviewee's involved in medical dispute authentic feelings. To be more specific, detected emotions could be approximately categorized in the Kübler-Ross Change Curve (KRCC), including grief denial, anger, bargaining, depression, acceptance, and coping strategies could be conducted according to individual differences. However, one thing should be kept in mind that this category is just an auxiliary measure for approaching the interviewer's authentic feelings. Stiffly putting someone's fragment of emotional journey into the framework of the KRCC is inappropriate³⁸. These detections of emotions by a well-educated mediator facilitate precise and rapid responses to such

feelings because physical discomfort leads to mental and emotional suffering³⁹. As physical health decline, either temporarily or permanently, would have mutual⁴⁰ and detrimental effects on psychological health (e.g., anxiety⁴¹, depression⁴²), leading to eventually undermining the existing/ established-in-the-process partnerships and difficulty in building trust with others⁴³. The two-hour practice in this segment of the TIME training program is designed to allow trainees to be proficient in providing emotional support through situational exercise and to receive verbal feedback on their performances by trainers.

When a medical dispute occurs, both medical personnel⁴⁴ and patients/relatives⁴⁵ would be influenced and become vulnerable, sensitive, and self-absorbed. The defense mechanism will be activated in both parties and cause the relationship to become increasingly alienated and destructive, and make people more injured, bringing about a form of vicious cycle. To break this cycle and restore the broken trust, the so-called 'transformative mediation'⁴⁶ with two key features is fundamental and would be practiced in the practicing part of the course section 'Conflict resolution'. One is empowerment shift⁴⁷, which could allow both parties to regain respect, confidence, and empathy. The other is recognition shift⁴⁷, which could help patients reduce their defensiveness against medical staff and gain a different perspective. This approach made patients express needs and strengthened bilateral participation in health care issue⁴⁸.

As we focused on the arm of patients/relatives, there would be ignorance of the need for care from the arm of medical personnel. This constantly neglected blind spot of calling for help for healthcare professions is going to be smoothed by mediators after the TIME training. The potential emotional trauma of staff in response to adverse patient responses can cause burnout⁴⁹. Such burnout has been thought to be high among physicians globally⁵⁰. Con-sequences of burnout have detrimental effects on doctors, which may lead to them making incorrect decisions (thus causing medical errors), exhibiting a hostile attitude towards patients, and having dysfunctional relationships with colleagues⁴⁹. Once staff having traumatic experiences they would suffer from second victim syndrome (SVS)⁵¹. Mediators after the TIME training could recognize the six phrases, including the early stage of 'initial chaos and accident response' and the final of 'moving on'⁵². In half of the process during the SVS recovery trajectory, especially in phrase three, the suffering personnel would try to seek help and re-establish trust with surroundings⁵². This would be the timing on target for mediators to approach involved personnel by listening empathetically, eschew judgement, and showing respect⁵³. SVS cannot be treated if systematic and well-structured support for coping with emotional burdens is absent⁵⁴. To put it in a nutshell, medical staff should be provided with adequate care similar to support for patients/relatives with mature, intact, and positive counselling⁵⁵.

Some limitations must be noted. First, the sample composition was imbalanced due to an uneven distribution of specialties caused by the method of purposive sampling. Doubts arise as any possibility of selection bias still left. Second, a retrospective follow-up to assess training results was lacking. To understand the quality of the mediation service under current circumstances, establishment of a distinctive measurement tool with indicators for capabilities of perception, accountabilities, communication, and assistance is recommended. Third, a feedback mechanism should set up for the collection of opinions and suggestions from onboard mediators, specialists taking part in any step of the mediation or litigation, or our greenhorns to revise the curriculum. This syllabus could be considered a starting point, but further research is required to achieve perfection.

Conclusions

The TIME syllabus is the first official training program of mediators for medical disputes in Taiwan and is constructed after a four-round Delphi process with four sections, including 'Knowledge base of law', 'Internalisation of the denotative and connotative meanings of care',

'Effective, smooth, and timely communication', and 'Conflict resolution'. The design of the TIME syllabus, which contains lectures and field exercises for trainees to enrich their knowledge in relation to mediation and sharpen their skills in establishing firm relationships with medical personnel and/or patients/relatives and managing medical disputes.

(09) References

(09-01) Reference 15: you have the author's first name, rather than last name in the list (should be Sumsion, T)

16. Sumsion T. The Delphi technique: an adaptive research tool. *Br J Occup Ther*;61(4):153-6.

(09-02) 40 also includes a first name.

47. Sally H. Transformative Mediation and Gestalt: The Good Form of Empowerment and Recognition: A Response to Lisa Gaynier. *Gestalt rev* 2003;7(3):204-9.

(09-03) Note that your citation of tables (references 19-21) will need formatting in house-style too.

2. Table of factual test of causation annually Taiwan: Ministry of Health and Welfare; 2021. Available from: <https://www.mohw.gov.tw/dl-67162-0dff04eb-1f24-4853-b49c-ce4959ed16b6.html>. (accessed 5 August 2021).

VERSION 2 – REVIEW

REVIEWER	Lin , Jin-Ding Mackay Medical College, Institute of Long Term Care
REVIEW RETURNED	09-Jul-2022
GENERAL COMMENTS	The author have revised the manuscript completely, I suggest it can be accepted for pulication without further revision.